# miR-21-5p Regulates the Proliferation and Differentiation of Skeletal Muscle Satellite Cells by Targeting KLF3 in Chicken

**DOI:** 10.3390/genes12060814

**Published:** 2021-05-26

**Authors:** Donghao Zhang, Jinshan Ran, Jingjing Li, Chunlin Yu, Zhifu Cui, Felix Kwame Amevor, Yan Wang, Xiaosong Jiang, Mohan Qiu, Huarui Du, Qing Zhu, Chaowu Yang, Yiping Liu

**Affiliations:** 1Animal Genetic Resources Exploration and Innovation Key Laboratory of Sichuan Province, Sichuan Agricultural University, 211 Huiming Road, Wenjiang, Chengdu 611130, China; 2019202006@stu.sicau.edu.cn (D.Z.); jsran0924@163.com (J.R.); jingjingyi11@126.com (J.L.); 2018102013@stu.sicau.edu.cn (Z.C.); amevorfelix@gmail.com (F.K.A.); as519723614@163.com (Y.W.); zhuqingsicau@163.com (Q.Z.); 2Breeding and Genetics Key Laboratory of Sichuan Province, Sichuan Animal Science Academy, 7 Niusha Road, Jinjiang, Chengdu 610066, China; yuchunlin1984@sina.com (C.Y.); xsjiang2017@163.com (X.J.); Mohan.qiu@163.com (M.Q.); duhuarui@sina.com (H.D.)

**Keywords:** myotubes, miR-21-5p, myogenic markers, muscle development

## Abstract

The proliferation and differentiation of skeletal muscle satellite cells (SMSCs) play an important role in the development of skeletal muscle. Our previous sequencing data showed that miR-21-5p is one of the most abundant miRNAs in chicken skeletal muscle. Therefore, in this study, the spatiotemporal expression of miR-21-5p and its effects on skeletal muscle development of chickens were explored using in vitro cultured SMSCs as a model. The results in this study showed that miR-21-5p was highly expressed in the skeletal muscle of chickens. The overexpression of miR-21-5p promoted the proliferation of SMSCs as evidenced by increased cell viability, increased cell number in the proliferative phase, and increased mRNA and protein expression of proliferation markers including PCNA, CDK2, and CCND1. Moreover, it was revealed that miR-21-5p promotes the formation of myotubes by modulating the expression of myogenic markers including MyoG, MyoD, and MyHC, whereas knockdown of miR-21-5p showed the opposite result. Gene prediction and dual fluorescence analysis confirmed that KLF3 was one of the direct target genes of miR-21-5p. We confirmed that, contrary to the function of miR-21-5p, KLF3 plays a negative role in the proliferation and differentiation of SMSCs. Si-KLF3 promotes cell number and proliferation activity, as well as the cell differentiation processes. Our results demonstrated that miR-21-5p promotes the proliferation and differentiation of SMSCs by targeting KLF3. Collectively, the results obtained in this study laid a foundation for exploring the mechanism through which miR-21-5p regulates SMSCs.

## 1. Introduction

Skeletal muscles are involved in a wide range of life activities, including thermoregulation, movement, organ protection, respiration, and buffering. In meat-producing animals, the development of skeletal muscle is closely correlated with the quality and quantity of meat products. The body of newly born animals is made up of numerous numbers of muscle fibers that are basically fixed together. Thus, myogenic stem cells and SMSCs provide myogenic precursors for rebuilding damaged muscle tissues and are involved in productive hypertrophy and muscle injury repair, which contribute to muscle production and maintenance [1,2]. In addition to major transcriptional factors, DNA methylation, non-coding RNA (ncRNAs), and signaling pathways also play significant regulatory roles in the growth and development of skeletal muscles, including myopathy, remodeling, growth, and repair [3,4,5,6].

MicroRNAs (miRNAs) are functional RNAs containing about 21 nucleotides that are formed from long transcripts after two consecutive splicing processes, which are essential for the development of an organism [7]. It was also reported to participate actively in the regulation of gene expression at the post-transcriptional level, resulting in mRNA degradation or translation disruption, by complete or incomplete complementary pairing with the 3′ UTR of target genes, or by binding its 5′ UTR and CDS regions [8,9]. These small non-coding RNAs are essential for regulating intracellular translation processes. In poultry, miRNAs are closely related to fat metabolism, muscle development, embryonic development, and disease control [10,11,12,13]. As far as we know, reports indicated that miRNAs, such as miR-206, miR-133, and miR-1, are highly expressed specifically in the muscular tissues and are regarded as key regulatory elements of cardiovascular development; hence, they are referred to as muscle-specific miRNAs (myomiRs) [14,15,16]. However, the effects of novel miRNAs on skeletal muscle formation and their regulatory mechanism require further studies. Our preliminary sequencing data showed that miR-21-5p was highly expressed in the muscle tissues of broilers during the embryonic period (E7, E11, and E17) and after birth (D1). miR-21 was first widely studied in human diseases, involving cardiovascular and cancer [17]. A Subsequent study reported that miR-21 plays an central role in hypoxia-induced proliferation and migration of pulmonary vascular smooth muscle cells [18]. The effect of miR-21 on skeletal muscle cells has also been reported in many animal models. For instance, miR-21 was expressed throughout the skeletal muscle development of Tongcheng pigs and was significantly (17-fold) differentially expressed in skeletal muscle at 90 days post coitus (E90) and 100 days postnatal (D100) [19]. Currently, studies focused on the effects of miR-21-5p on vascular smooth muscle and cardiac muscle. However, its effect on skeletal muscle formation and its regulatory mechanism requires further study.

Based on miR-21-5p target gene prediction and RNA sequencing results, we identified Krüppel-like factor 3 (KLF3) as a candidate target gene of miR-21-5p in chicken skeletal muscle satellite cells. KLF3 is one of the founding members of Krüppel-like factor (KLFs), a subfamily of zinc finger-like DNA-bound transcription regulators capable of both gene trans activation and gene suppression [20]. In skeletal muscle biogenesis, KLFs are involved in myogenesis and muscle maturation, regulation of postpartum muscle plasticity, integrated metabolic function, and muscle diseases [21]. KLF10 interacts with FGFR1 promoter to inhibit proliferation and regulate differentiation of myoblasts during muscle formation in chicken [22]. As a transcriptional regulator, KLF3 can be enriched at the MPEX site of the MCK promoter and many other muscle gene promoters and both KLF3 subtypes are up-regulated during muscle differentiation [23]. In addition, KLF3 expression is initiated late in muscle differentiation, when many of the genes that define mature myotubes are induced and it may play a role in gene suppression later in muscle development [21]. A recent study identified a previously unrecognized heterogeneity of KLF3, which promotes 8-cell-like transcriptional status in pluripotent stem cells [24]. Although the molecular mechanisms by which KLF3 regulates transcription have been extensively analyzed and established, its role in skeletal muscle proliferation and differentiation has not been fully understood in animal models including poultry. Therefore, in this study, we investigated the role of miR-21-5p in proliferation and differentiation of chicken SMSCs and explored its mechanism of action. We further demonstrated the reverse inhibitory effect of KLF3 on the proliferation and differentiation of SMSCs.

## 2. Materials and Methods

### 2.1. Ethics Standards

The procedures followed in the animal experiment of this study were approved by the Animal Welfare Committee of Sichuan Agricultural University, Chengdu, China (ZDH-20192020062). All experimental steps were performed in accordance with relevant guidelines and regulatory requirements.

### 2.2. Animals and Samples

Six Daheng broilers of 3 days old were selected. Samples including the heart, liver, spleen, lung, kidney, breast muscle, leg muscle, muscle stomach, glandular stomach, small intestine, and brain were collected, wrapped in a tin aluminum foil, and immediately frozen in liquid nitrogen and later stored at −80 °C for further analyses.

### 2.3. Cell Culture

Following the methods described previously by Bai et al. [25], SMSCs were isolated and cultured from the breast muscle of 3-day-old Daheng broilers. After the primary cells were collected, the SMSCs were only used for one generation in each culture. The pectoral muscle was collected and shredded to release cells with 0.1% collagenase I (Sigma Chemical Co., St. Louis, MO, USA) and 0.25% trypsin (Gibco, Grand Island, NY, USA). The satellite cells were isolated from the cell suspension by filtration and differential adhesion. Then 10% growth medium (GM: Dulbecco’s modified Eagle medium (DMEM) (Gibco, Grand Island, NY, USA), +10% fetal bovine serum (Gibco, Grand Island, NY, USA), +0.2% penicillin/streptomycin (Invitrogen, Carlsbad, CA, USA)) was added to culture the isolated satellite cells. When the cell density reached about 70–80% in the growth medium, the cells were cultured in differentiation medium (DM: DMEM + 2% horse serum (Gibco, Grand Island, NY, USA) instead, which is used to induce differentiation. The cells were cultured in a cell incubator at a constant temperature and humidity (Thermo Scientific, San Jose, CA, USA) (5% CO_2_ humid atmosphere, 37 °C) and then medium was changed daily. The cells were collected from the growth medium (GM) and differentiation medium at 24, 36, 48, and 72 h (DM1, DM2, DM3, and DM4). DF-1 cells were used for dual-luciferase reporter assay and cultured in 10% GM and medium was also replaced every 24 h. At each stage, biological replicates of 3 cell wells were collected.

### 2.4. RNA Oligonucleotides, Vectors, and Transfection

miR-21-5p mimic, mimic negative control (mimic NC), miR-21-5p inhibitor, inhibitor NC, and KLF3 small interfering RNAs (siRNAs) were synthesized by GenePharma Co., Ltd. (Shanghai, China). Wild-type and mutated sequences of miR-21-5p binding site in the 3′UTR region of KLF3 were synthesized and constructed into pmirGLO vectors (Promega, Madison, WI, USA) using NheI and XhoI restriction sites according to the instructions. Oligonucleotide sequences are provided in Table 1.

According to the manufacturer’s instructions, SMSCs were transfected with lipofectamine 3000 reagent (Invitrogen, Carlsbad, CA, USA) and the medium was replaced 6–8 h later. After the about 50–60% of the cells were cultured in GM, they were transfected for cell proliferation-related tests. Thereafter, for cell differentiation, a fusion rate of about 80–90% was obtained and then transfected in DM.

### 2.5. RNA Isolation, Complementary DNA (cDNA) Synthesis, and Real-Time Quantitative PCR (qRT-PCR)

Total RNA was extracted from tissues or cells using TRIzol Reagent (Takara, Dalian, China) under the guidance of the manufacturer. The sample RNA concentration and the integrity was determined by measuring the optical density of the sample with Thermo Scientific^™^ NanoDrop Lite (Thermo, Waltham, MA, USA) and the concentration was adjusted to be consistent. mRNA reverse transcription was completed with PrimeScript RT Master Mix Perfect Real Time (Takara, Dalian, China), while reverse transcription reactions for miRNA were performed using the One Step miRNA cDNA Synthesis Kit, as per manufacturer’s instructions (HaiGene, Haerbin, China). qRT-PCR analysis was performed in 10 µL reaction volumes containing 1 µL cDNA, 0.5 µL forward and reverse primers, 5 µL TB Green^™^ premix (Takara) and 3 µL DNase/RNase Free deionized water (Tiangen, Beijing, China). All amplicon primers sets were designed in the National Center for Biotechnology Information (NCBI) database and synthesized by Sangon Biotech Primer Design Center (Shanghai, China); details of the primers used are shown in Table 2. The 2^−∆∆Ct^ method was used to analyze the relative expression level of different qRT-PCR data [26]. The reference genes in mRNA and miRNA quantification were GAPDH and U6, respectively, which were relatively constant in cells and did not change under treatment conditions.

### 2.6. Protein Extraction and Western Blot Analysis

The proteins were extracted on ice using commercial protein extraction kits (BestBio Biotech Co., Ltd., Shanghai, China) and adjusted to the same concentration, then placed at 95 °C to denature for five minutes. The total volume of each cell well was 20 µL, including 16 µL protein sample and 4 µL reducing loading buffer (4:1). The steps and details of the Western Blot Analysis experiment are described in depth by Cui et al. [27]. The primary antibodies were diluted, according to the manufacturer’s instructions as follows: myosin heavy chain, cardiac muscle complex (MyHC, cat. no. sc-32732, Santa Cruz Biotechnology, CA, USA; 1:200), myogenin (MyoG; Biorbyt, Cambridge, UK; diluted, 1:1000), cyclin-dependent kinase 2 (CDK2; ABclonal Technology, Wuhan, China, 1:2000), proliferating cell nuclear antigen (PCNA; ABclonal, 1:5000), and KLF3 (ABclonal, Wuhan, China; 1:1000), and the β-tubulin (ZenBio, Chengdu, China; 1:2000) was used as a loading control. The secondary antibody are as follows: HRP Goat Anti-Rabbit IgG (H + L) (ABclonal, Wuhan, China; 1:5000); Goat Anti-mouse IgG (Biorbyt, Cambridge, UK; diluted, 1:5000).

When the density of transfected SMSCs reached about 60% and in logarithmic growth phase, cell proliferation was detected with C10310-1/-2/-3 EdU Apollo in vitro imaging kit (RiboBio, Guangzhou, China). Cells growing in 96-well plates were added with 100 µL and 50 µM EdU per well and then cultured in cell incubator for 3 h. The cells were then thoroughly washed with phosphate buffered saline (PBS) and fixed with 4% paraformaldehyde for 30 min. After adding 50 µL of 2 mg/mL Glycine and 100 µL of 0.5% Triton X-100 PBS, respectively, the cells were washed with PBS. Afterwards, 100 ul of Apollo staining reaction solution was added to each well and were incubated in dark for 0.5 h. Furthermore, the cell nuclei were counter-stained with Hoechst-33342 for 0.5 h. After the final incubation, fluorescence microscope (Olympus, Tokyo, Japan) was used to randomly select 3 fields for observation and photography of stained cells. The images and data were processed using Image-Pro Plus software.

### 2.7. Cell Counting Kit 8 (CCK-8) Assay

SMSCs were seeded in 96-well plates and transfected with siRNA, negative control, mimics, or inhibitors. A Cell Counting Kit-8 kit (Multisciences, Hangzhou, China) was used to detect cell proliferative activity, according to the manufacturer’s instructions. Then, 10 ul CCK-8 Reagent was added to each cell well at 12 h, 24 h, 36 h, and 48 h after cell transfection and the cells were cultured in an incubator at a constant temperature (37 °C, 5% CO_2_) for 2 h. The optical density (OD) of each sample at 450 nm was measured by Thermo Scientific^™^ Varioskan LUX at 450 nm.

### 2.8. Immunofluorescence Assay

SMSCs were seeded in 12-well plates and cultured in DM and transfected. After the cells grew and differentiated into myotubes, they were fixed with 4% formaldehyde for 20 min and washed with PBS for 3 consecutive times for 5 min. Subsequently, 0.1% Triton X-100 was added and cultured for 20 min to promote cell permeability, followed by the addition of 5% goat serum (Beyotime, Shanghai, China) for 30 min to block the cells. Furthermore, the primary antibody (MyHC; Santa Cruz; 1:250) was added and incubated overnight at 4 ^◦^C. After that Rhodamine (TRITC) AffiniPure Goat Anti-Mouse Immunoglobulin G (IgG; ZenBio; 1:1000) was added and the cells were incubated at room temperature for 1 h. Then the cells were rinsed with PBST for three times, thereafter, 4′, 6-diamidino-2-phenylindole (DAPI; Beyotime; 1:50) was added for nuclear staining. Five minutes after staining, the anti-fluorescence quencher (Beyotime) was added and observed under a fluorescence microscope (Olympus, Tokyo, Japan). The mean number of nuclei in each myotubes is equal to the total number of nuclei of the myotubes divided by the total number of myotubes. Three random images were taken from each well and the area, diameter, and average nuclei number of the myotube were measured using Image-Pro Plus software.

### 2.9. Luciferase Reporter Assay

DF-1 cell lines of chicken embryo fibroblast cells (DF-1 cells) were seeded in 48-well plates. DF-1 cells were cultured with DMEM + 10% fetal bovine serum and when the cell density reached about 70–80%, they were co-transfected with miR-21-5p mimic or mimic NC and KLF3-wild-type (WT) or KLF3-mutant (MT) plasmid, with three replicates for each treatment. Then, after 48 h of transfection, the cells were treated with Dual-GLO Luciferase Assay System Kit (Promega), as per the manufacturer’s instructions, and then the luciferase activity of firefly luciferase and renal cell luciferase was measured using a Fluorescence/Multi-Detection Microplate Reader (Biotek, Shoreline, WA, USA).

### 2.10. Bioinformatic Analysis

miRNA target gene prediction was performed by TargetScan website (http://www.targetscan.org/vert_71/) (accessed on 5 January 2018), miRDB website (http://mirdb.org/index.html) (accessed on 22 March 2019) and DIANA website (http://diana.imis.athena-innovation.gr/DianaTools/index.php?r=microT_CDS/index) (accessed on 8 May 2019). Venn analysis was performed by Venn diagrams website (http://bioinformatics.psb.ugent.be/webtools/Venn/) (accessed on 22 March 2019).

### 2.11. Statistical Analysis

The data collected were analyzed using SPSS20.0 statistical software. A one-way ANOVA was used for multiple-group comparison analysis [28]. Differences were considered significant at the *p*-value < 0.05 (*) or *p*-value < 0.01 (**) and different lowercase letters above bars indicate significant differences (*p* < 0.05). All data were derived from at least three replicates of experimental processing and data are presented as means ± standard error of the mean (SEM).

## 3. Results

### 3.1. Expression of miR-21-5p in Chickens

Sequence alignment analysis showed that miR-21-5p was highly conserved and there was almost no difference among different species (Figure 1A). Previous RNA-seq showed that the miRNAs with the highest total expression levels at four time points in the embryonic period included miR-21-5p (Figure 1B). In addition, both miR-1a-5p [29] and miR-148-3p [30] are critical transcriptions related to the growth and development of skeletal muscle. The results on the expression of miR-21-5p in the skeletal muscle and other tissues showed that miR-21-5p was highly expressed in the breast muscle and leg muscle higher than those expressed in the heart, lung, gizzard, and other tissues, whereas the lowest expression was observed in the brain and glandular stomach (Figure 1C). To investigate the expression pattern of miR-21-5p during the development of skeletal muscle in chickens, we conducted qRT-PCR analysis and the results showed that the expression level of miR-21-5p gradually increased during the development of skeletal muscle in chicken embryos (Figure 1D). The above results indicated that miR-21-5p may play a role in the growth and development of skeletal muscle in poultry.

### 3.2. miR-21-5p Promotes the Proliferation of Chicken SMSCs

qRT-PCR, CCK-8, Western Blot Analysis, and EdU assays were used to investigate whether miR-21-5p plays a role in the proliferation of SMSCs. We transfected the synthesized miR-21-5p mimic, mimic NC, miR-21-5p inhibitor, and inhibitor NC into cells and the transfection efficiency is shown in Figure 2A,B (*p* < 0.01), indicating that it was successfully down-regulated or overexpressed. In parallel, the expression of cell proliferation marker genes PCNA, CCND1, and CDK2 were detected. qRT-PCR results showed that the overexpression of miR-21-5p increased the expression levels of PCNA, CCND1, and CDK2. Conversely, their expression levels significantly decreased after miR-21-5p knockdown, except for CDK2 (*p* < 0.01; Figure 2C,D). Furthermore, the Western Blot results showed that the expression of PCNA and CDK2 was promoted by transfection with miR-21-5p mimics, whereas it was inhibited by transfection with miR-21-5p inhibitor (*p* < 0.05; Figure 2E–G). CCK8 assay was performed to detect cell viability and the results showed that SMSCs proliferation was significantly inhibited following miR-21-5p knockdown (*p* < 0.05; Figure 2H,I). Meanwhile, the proliferation ability of SMSCs was detected by EdU assay and the results showed that as the miR-21-5p was overexpressed, the proportion of EdU positive cells increased (*p* < 0.01; Figure 2J–M). These results suggested that miR-21-5p may promote the proliferation of SMSCs.

### 3.3. miR-21-5p Promotes the Differentiation of Chicken SMSCs

In order to investigate the effect of miR-21-5p on the differentiation of chicken SMSCs, the SMSCs were induced to differentiate in vitro when they had reached 70–80% confluency. The qRT-PCR results show that the expression level of miR-21-5p increased gradually during differentiation (*p* < 0.01; Figure 3A). Meanwhile, the mRNA and protein expressions of intracellular differentiation markers MyoG and MyHC were detected in the experimental group and control group respectively after the overexpression and knockdown of miR-21-5p, as well as the formation of myotubes. The results showed that the expression of miR-21-5p was increased in the differentiated SMSCs and the relative expression of MyoG and MyHC mRNAs were significantly increased compared with that in the control group (*p* < 0.05; Figure 3B,C). Similarly, the Western Blot experiments indicated that the MyHC and MyoG proteins were up-regulated after miR-21-5p overexpression, while they were down-regulated after miR-21-5p inhibition (*p* < 0.05; Figure 3D–F). In addition, immunofluorescence assay showed a significant increase in myotubes area, nuclei on average and myotube diameter after overexpression of miR-21-5p, whereas the opposite was true for SMSCs treated with miR-21-5p inhibitors (*p* < 0.05; Figure 3G–N). Collectively, these results elucidated that miR-21-5p promoted the differentiation of chicken SMSCs.

### 3.4. miR-21-5p Targets Directly KLF3 Gene

We further investigated the mechanisms through which miR-21-5p regulated proliferation and differentiation of SMSCS by using three online software tools (TargetScan, miRDB and DIANA) combined with Venn’s analysis to predict the target genes. The results showed that 16 target genes were found on each of the three online websites (Figure 4A). Since many members of the KLFs family play a pivotal role in the proliferation and differentiation of skeletal muscle [31,32], KLF3 was chosen for subsequent validation. The direct target relationship between miR-21-5p and KLF3 in chicken SMSCs was verified with dual luciferase reporter experiment. The 3′UTR target sequence was cloned into the luciferase reporter vector (pmirGLO-KLF3-3′UTR WT) and a luciferase reporter vector was constructed that mutated the target sequence binding site (pmirGLO-KLF3-3′UTR MT) (Figure 4B,C). Then, DNA-seq results showed that luciferase reporter vectors were successfully constructed and could be used in subsequent experiments (Figure 4D). When DF-1 cells were co-transfected with miR-21-5p mimics and pmirGLO-KLF3-3′UTR, luciferase activity was decreased. However, no changes were detected in DF-1 cells co-transfected with the mutant reporter gene (*p* < 0.01; Figure 4E). Subsequently, we also detected the mRNA and protein levels of KLF3 in SMSCs after miR-21-5p overexpression and knockdown. The results showed that overexpression of miR-21-5p decreased KLF3 mRNA and protein levels, while inhibition of miR-21-5p increased the level of KLF3 mRNA (*p* < 0.05; Figure 4F,G). These results sufficiently demonstrated that there is a direct target region relationship between miR-21-5p and KLF3.

### 3.5. Knockdown of KLF3 Facilitates Chicken SMSCs Proliferation

Based on the previous predictions, we investigated the expression of KLF3 in different tissues of chickens and the results showed that KLF3 expression was highest in breast muscle and leg muscle, as well as the expression of miR-21-5p (Figure 5A). To determine the potential roles of KLF3 in the proliferation and differentiation of chicken SMSCs, three different siRNA interference vectors were transfected into the chicken SMSCs (*p* < 0.01; Figure 5B). The knockdown efficiency of si-KLF3-77 on KLF3 protein level was also determined by Western blotting, with the results showing that si-KLF3-77 significantly decreased the protein level of KLF3 (Figure 5C). Thus, si-KLF3-77 was chosen for subsequent experiments.

As expected, qRT-PCR showed that KLF3 knockdown significantly upregulated the expression levels of proliferation-related genes in PCNA, CDK2, and CCDN1 in SMSCs (*p* < 0.01; Figure 5D). Similarly, protein levels of CDK2 and PCNA increased significantly after KLF3 was successfully interfered (*p* < 0.01; Figure 5F,G). In addition, after SMSCs cells were treated with CCK-8 reagent, the optical density of si-KLF3 group decreased significantly, which represented the proliferation state of cells (*p* < 0.01; Figure 5E). Furthermore, the proportion of positive EdU cells increased after lowering the expression of KLF3 (*p* < 0.01; Figure 5H,I). This indicated that KLF3 has a negative effect on the proliferation of chicken SMSCs.

### 3.6. Knockdown of KLF3 Facilitates Chicken SMSCs Differentiation

qRT-PCR results suggested that KLF3 may play a potential role in the differentiation of SMSCs, as its expression is significantly reduced after SMSCs began to differentiate (*p* < 0.01; Figure 6A). Meanwhile, the mRNA expression of MyoD, MyoG, and MyHC was upregulated by KLF3 interference (*p* < 0.01; Figure 6B). Similarly, MyHC and MyoG protein levels were inhibited by KLF3 and upregulated after KLF3 interference (*p* < 0.05; Figure 6C,D). Moreover, these results were confirmed by immunofluorescence analysis. After KLF3 knockdown, the area of the myotubes changed and KLF3 knockdown promoted the formation of the myotubes (*p* < 0.05; Figure 6E,F). In summary, these results suggest that KLF3 may be an inhibitor of SMSCs differentiation and myotubes formation.

## 4. Discussion

Chickens are key scientific models for studying skeletal muscle development in animals, because the development of chicken anatomy mimics that of mammals. Cell lineage studies, combined with cell imaging techniques, allow biological phenomena such as cell proliferation and differentiation to be easily observed and hypotheses to be tested extremely rapidly-a strength limited to avian systems. [33,34]. Skeletal muscle is one of the most abundant tissues in vertebrates. Abnormal development of skeletal muscle may lead to contractile injury and metabolic and endocrine abnormalities [35]. Currently, the development of chicken genome sequencing and new molecular techniques has identified many marker genes and regulatory genes.

In the present study, we aimed to explore the role and mechanism of miR-21-5p and its target genes in skeletal muscle development. Data from previous and current transcriptome studies demonstrated that miR-21-5p promotes proliferation of chicken SMSCs and accelerates cell differentiation and fusion into myotubes. The promotion effect of miR-21-5p on the proliferation and differentiation of SMSCs was mediated by KLF3.

The proliferation and differentiation of muscle cells are two key processes that determine the development of skeletal muscle. Importantly, miRNA is implicated in skeletal muscle formation. In addition to myomiRs, some broad-spectrum miRNAs, such as miR-543 [31], miR-7 [32], miR-199b [36], miR-146b [37], miR-181 [38], etc., are also involved in skeletal muscle development, among them, miR-7 and miR-146b were verified with chicken primary muscle cells. In studies employing vertebrate models of development, miR-21 expression has been detected at very early stages of development [39]. In other species, miR-21 has been shown to promote cell proliferation and inhibit apoptosis [40]. In addition, miR-21 also promotes osteogenic differentiation of mesenchymal stem cells (MSCs), which are critical for bone repair, through the PI3K/β-catenin pathway [41]. At present, the most commonly used verification methods are fluorescence assay of marker genes on mRNA level, protein hybridization assay and detection of cellular level and metabolic pathway. These techniques were also used in this study. In the current study, we found that the overexpression of miR-21-5p could increase the expression of proliferation marker genes (CDK2, PCNA, and CCND1) in SMSCs and increase the proliferation rate and activity of SMSCs. Studies have shown that miR-21 can target PTEN and FASLG to promote liver regeneration and hepatocyte proliferation in mouse, which also confirms that miR-21 is beneficial to cell proliferation [42,43]. These results are consistent with ours, suggesting that miR-21-5p is involved in the regulation of cell proliferation. Besides, miR-21 has been shown to promote osteogenic differentiation; however, no study reported its role in skeletal muscle cells [44,45]. Meanwhile, in this study, miR-21-5p mimics promoted the differentiation of SMSCs in chicken by upregulating the levels of MyHC and MyoG, which was consistent with the differentiation markers used in previous reports [46]. In addition, immunofluorescence confirmed that miR-21-5p mimics promoted myotubes formation, while miR-21-5p inhibitors suppressed it.

miRNAs perform different biological functions via targeting different genes by inhibiting mRNA translation or triggering its degradation [47]. In order to explore which genes were targeted by miR-21-5p in regulating the proliferation and differentiation process of SMSCs, we used three target gene prediction software for Venn analysis and a total of 16 candidate genes were screened out. Interestingly, among the genes screened, KLF3, SOX5, MKX, and ARHGAP24 were the most biologically important transcripts and hence, were given much attention. SOX5 and MKX are closely related to muscle development and MKX can inhibit the expression of MyoD promoter by directly binding to it, which plays a negative regulatory role in muscle differentiation [48,49]. ARHGAP24 was reported to inhibit cell cycle progression, induces apoptosis, and suppresses invasion in renal cell carcinoma [50]. However, through preliminary experiments, only KLF3 expression was significantly decreased after miR-21-5p overexpression. It has been found that KLF3 acts synergistically with Serum Response Factor (SRF) to regulate muscle-specific gene expression [23]. Bioinformatics software and luciferase reporter assay were used to confirm that miR-21-5p showed a strong target relationship with KLF3. After the overexpression of miR-21-5p, the mRNA and protein expression levels of SMSCs were significantly down-regulated. Furthermore, previous reports have highlighted that KLF3 is one of the three novel target genes of miR-21 newly identified to be involved in hair follicle development in sheep [51]. Therefore, we speculate that KLF3 is the direct target gene of miR-21-5p that play roles in the proliferation and differentiation of SMSCs, but this function of KLF3 needs to be further verified.

Several KLFs are involved in the regulation of skeletal muscle development and other function in humans and animals. For instance, KLF2, KLF3, and KLF4 are involved in myogenesis and muscle maturation, KLF5 and KLF15 are involved in postpartum muscle plasticity and comprehensive metabolic function, moreover, KLF15 is also a regulator of energy metabolism in muscle tissue [21]. However, whether KLF3 has a regulatory effect on chicken SMSCs is unknown. Here, we constructed an experimental model on the proliferation and differentiation of SMSCs after KLF3 knockdown. The results of proliferation showed that KLF3 knockdown could promote cell viability of SMSCs and the proportion of SMSCs in proliferative phase. Similarly, KLF3 interference promoted the expression of differentiation-related proteins and the formation of myotubes during cell differentiation. Report indicated that the primary mechanism of action of KLFs is that they bind to CACCC elements and GC-rich DNA regions to mediate transcription activation and/or inhibition [52]. Although, a previous study indicated that KLF3 can act as a weak transcriptional activator [53], however, in most cases, it forms a powerful transcriptional silencing complex by recruiting the cosuppressor CtBP [54]. For instance, KLF3 inhibits Wee1 gene expression in mice by interacting with multiple upstream elements [55], whereas KLF8 possess two promoters containing multiple CACCC elements, however, KLF3 inhibits these two promoters to deactivate KLF8 [56]. Therefore, we hypothesized that the inhibitory effect of KLF3 on the development of SMSCs was attributed to its role as a transcriptional repressor through binding to specific domains or promoters of related genes. However, its specific mechanism requires further elucidation.

In conclusion, this study reveals a new mechanism by which miR-21-5p regulates the growth and development of skeletal muscle. It was established in this study that miR-21-5p regulates myogenesis by promoting proliferation and differentiation of SMSCs. Results obtained in this study also showed that there is a direct target region relationship between miR-21-5p and KLF3 (Figure 7).

## Figures and Tables

**Figure 1 genes-12-00814-f001:**
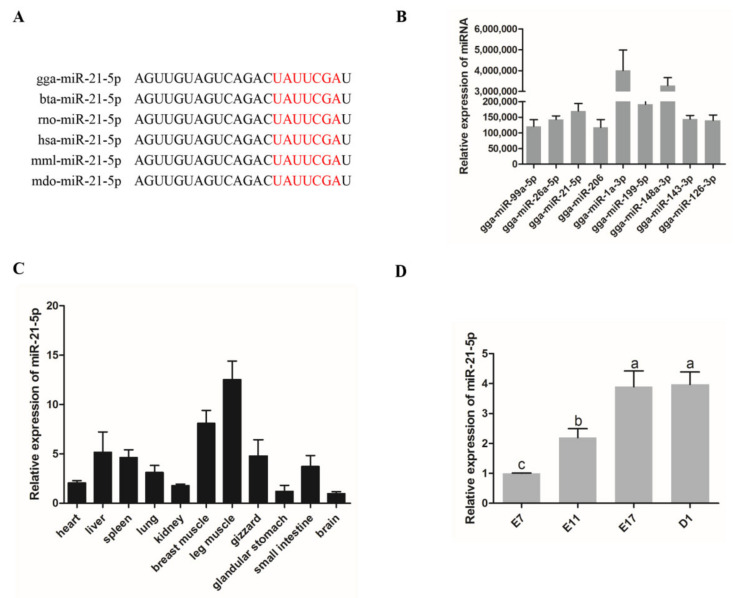
Expression of miR-99a-5p in chickens. (**A**) miR-21-5p sequence of different species: gga, chicken; bta, cattle; rno, rat; hsa, homo sapiens; mml, macaca mulatta; mdo, monodelphis domestica. (**B**) The miRNAs with the highest expression in embryoni (E7, E11, E17 and D1) chicken skeletal muscle. (**C**) Expression of miR-21-5p in different tissues of 3-day-old chickens. (**D**) Expression of miR-21-5p in skeletal muscle of chicken at four embryonic stages. The results were expressed as mean ± SEM. (*n* = 3). Different lowercase letters above bars indicate significant differences (*p* < 0.05). miR, microRNA.

**Figure 2 genes-12-00814-f002:**
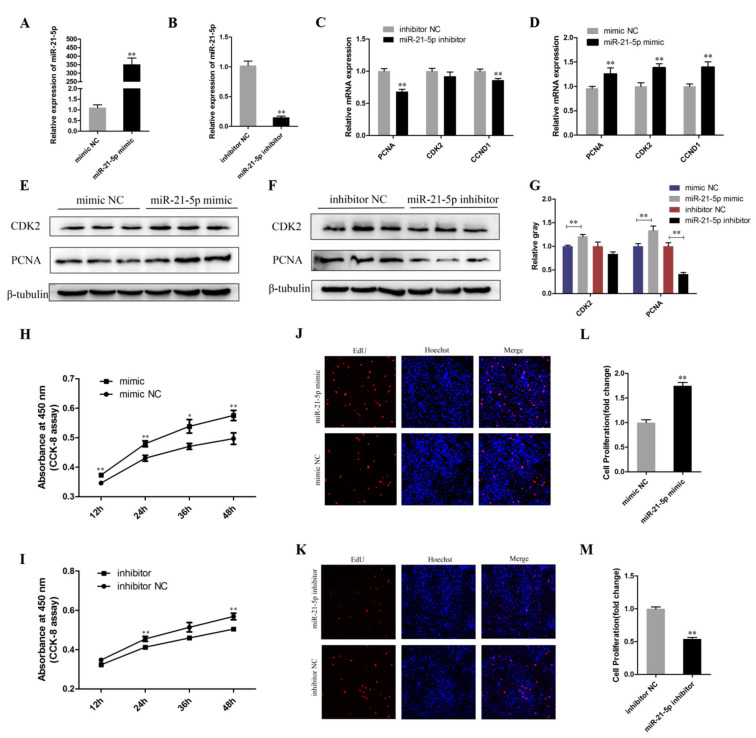
miR-21-5p promotes the proliferation of chicken SMSCs. (**A**,**B**) After transfection with miR-21-5p inhibitor and mimic, the expression of miR-21-5p in SMSCs was determined by using qRT-PCR. (**C**,**D**) The mRNA expression of PCNA, CDK2, and CCND1 after 24 h transfection of miR-21-5p mimic or inhibitor in SMSCs cells. (**E**–**G**) The protein expression of PCNA and CDK2 after 48 h transfection of miR-21-5p inhibitors and miR-21-5p mimics in SMSCs determined by Western blot analysis. β-Tubulin was used as a reference gene. (**H**,**I**) Cell viability was measured using the cell counting kit-8 (CCK-8). (**J**–**M**) 5-ethynyl-20-deoxyuridine (EdU) staining of transfected SMSCs and the calculation of the proliferation rate. The results were expressed as mean ± SEM. (*n* = 3). * *p* < 0.05 and ** *p* < 0.01.

**Figure 3 genes-12-00814-f003:**
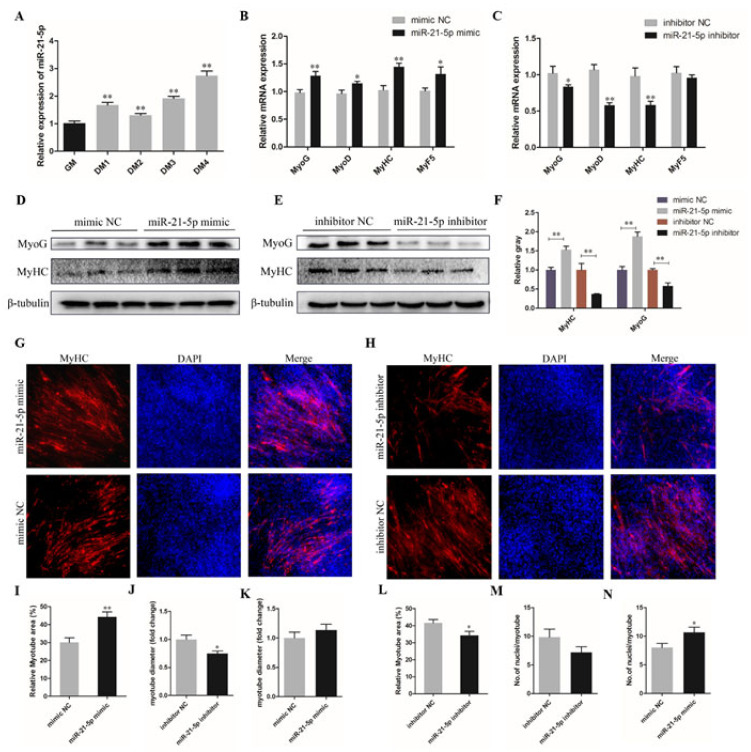
miR-21-5p promotes the differentiation of chicken SMSCs. (**A**) The expression level of miR-21-5p during proliferation (GM) and differentiation of SMSCs. DM24, DM36, DM48, and DM72 represent SMSCs which were induced to differentiate for 24, 36, 48, and 72 h, respectively. (**B**,**C**) The mRNA expression levels of MyoG, MyoD, MyHC, and MyF5 after 24 h of overexpression and inhibition of miR-21-5p in SMSCs. (**D**–**F**) The protein expression of MyoG and MyHC after 48 h of transfection of miR-21-5p mimic and inhibitor in SMSCs using Western Blot. (**G**,**H**) Anti-Myosin heavy chain (MyHC) immunofluorescence staining after the transfection of miR-21-5p mimic and inhibitor in SMSCs. DAPI (blue), cell nuclei; Merge: the fusion of SMSCs into primary myotubes. (**I,L**) Relative myotube area of chicken SMSCs following miR-21-5p overexpression and inhibition. (**J**,**M**) Immunofluorescent staining for MyHC in SMSCs myotubes showed that the average number of nuclei per myotubes and (**K**,**N**) Myotube diameter. The results were expressed as mean ± SEM. (*n* = 3). * *p* < 0.05; ** *p* < 0.01.

**Figure 4 genes-12-00814-f004:**
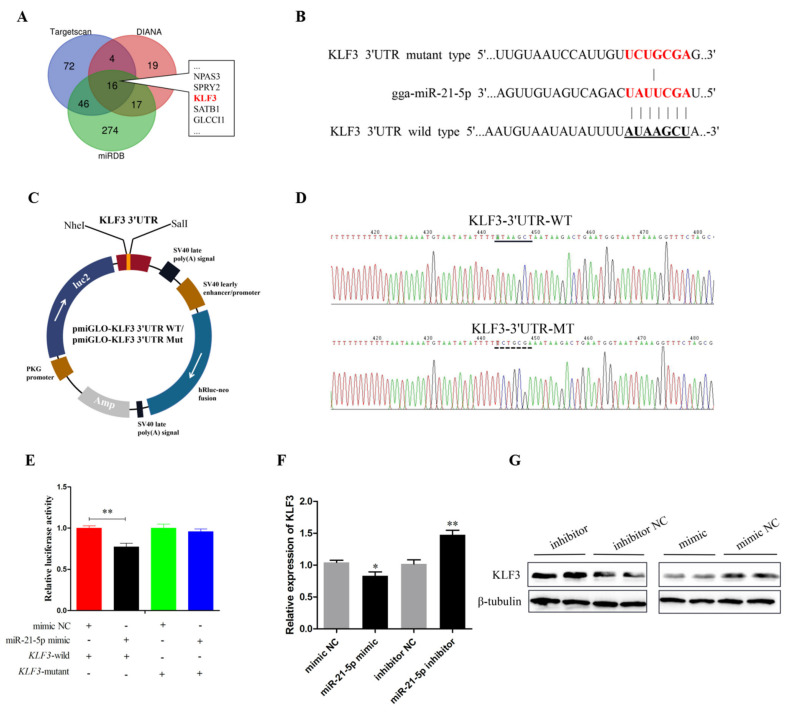
miR-21-5p directly targets KLF3 Gene. (**A**) Prediction of target genes of gga-miR-21-5p using DIANA, TargetScan, and miRDB. (**B**) The complementary pairing of gga-miR-21-5p with the targeted gene KLF3 3′UTR or mutated UTR. (**C**) Diagram of the construction of dual-luciferase reporter vectors containing the wild or mutant KLF3 3′-UTR sequences. hRluc means renilla luciferase; hluc2 means firefly luciferase. (**D**) Validation of wild-type plasmid and mutant plasmid. WT means wild-type vector; Mut represents mutant vector. (**E**) Chicken DF-1 cells were co-transfected with KLF3-3′-UTR wild or mutant dual-luciferase vector and the miR-21-5p mimic or mimic NC. The relative luciferase activity was assayed 48 h later. (**F**,**G**) After transfection with miR-21-5p mimics, miR-21-5p inhibitors or NC, the expression of KLF3 was determined by q-PCR and Western blot. The results were expressed as mean ± SEM. (*n* = 3). * *p* < 0.05; ** *p* < 0.01.

**Figure 5 genes-12-00814-f005:**
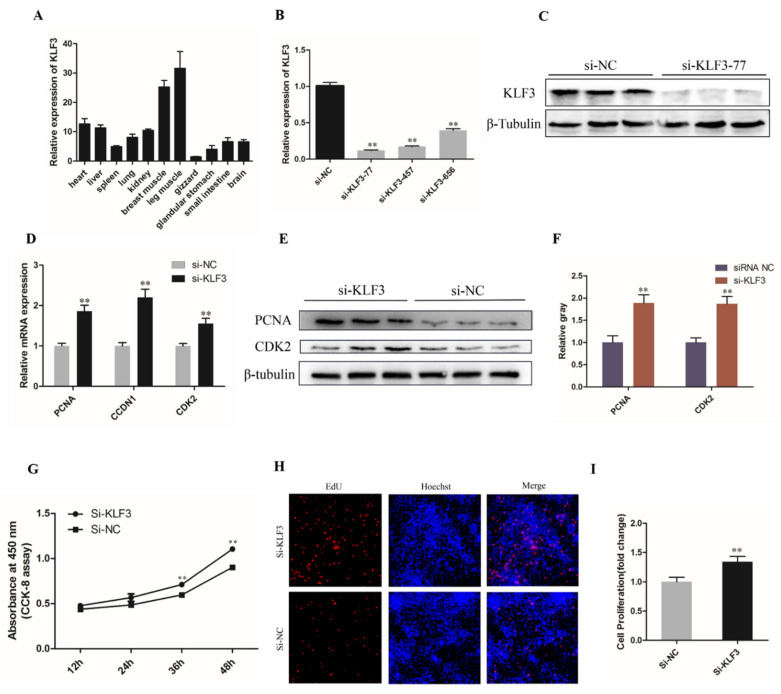
Knockdown of KLF3 facilitates chicken SMSCs proliferation. (**A**) Relative expression of KLF3 in different chicken tissues. (**B**) The knockdown efficiency of KLF3 gene in SMSCs by three small siRNAs were detected by qRT-PCR. (**C**) The protein expression level of KLF3 after interference by si-KLF3-77 was detected by Western blotting. (**D**) The mRNA expression of CDK2, PCNA, and CCND1 after 24 h of KLF3 knockdown. (**E**) CCK-8 assays for SMSCs after KLF3 knockdown. (**F**,**G**) The protein expressions of MyHC and MyoG after 48h of inhibition of the KLF3 gene in SMSCs using Western Blot. (**H**) Results of EdU assay for SMSCs after inhibition of KLF3 for 48 h, where EdU (red) fluorescence is used as an indicator of proliferation and nuclei are indicated by Hochest (blue) fluorescence. (**I**) The quantitative data of proliferating PSCs number in panel G. The results were expressed as mean ± SEM. (*n* = 3). ** *p* < 0.01.

**Figure 6 genes-12-00814-f006:**
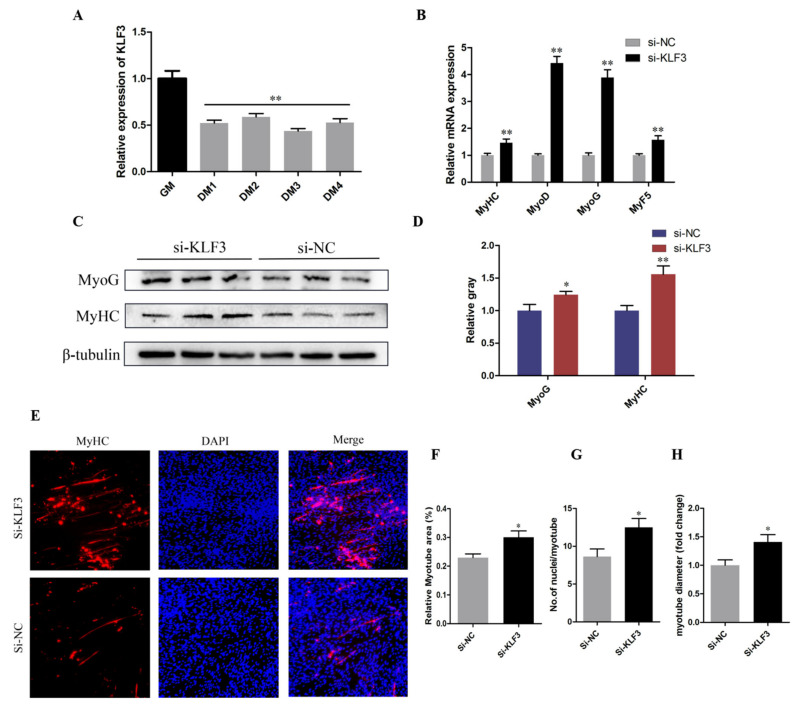
Knockdown of KLF3 facilitates chicken SMSCs differentiation. (**A**) The expression level of KLF3 during the proliferation (GM) and differentiation of SMSCs. DM24, DM36, DM48, and DM72 represent SMSCs which were induced to differentiate for 24, 36, 48, and 72 h, respectively. (**B**) The mRNA expression levels of MyoD, MyHC, MyoG, and MyF5 after 24h knockdown of KLF3 in SMSCs cells. (**C**,**D**) The protein expression levels of MyHC and MyoG after 48 h inhibition of KLF3. (**E**) Immunofluorescence analysis of SMSCs after KLF3 inhibition. (**F**) Relative myotube area of chicken SMSCs following KLF3 knockdown. (**G**,**H**) Immunofluorescent staining for MyHC in SMSCs myotubes showed that the average number of nuclei per myotubes and Myotube diameter. The results were expressed as mean ± SEM. (*n* = 3). * *p* < 0.05; ** *p* < 0.01.

**Figure 7 genes-12-00814-f007:**
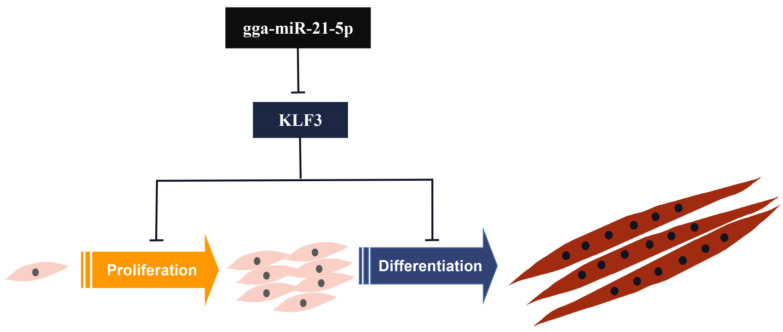
Schematic diagram of miR-21-5p mediated proliferation and differentiation of chicken skeletal muscle satellite cells.

**Table 1 genes-12-00814-t001:** RNA oligonucleotides in this study.

Fragment Name	Sequences (5′-3′)
KLF3 siRNA-77	F:GACCAAAUGAAGCCAAACATT R:UGUUUGGCUUCAUUUGGUCTT
KLF3 siRNA-457	F:GCCAGUUCCUUUCAUGUAUTT R: AUACAUGAAAGGAACUGGCTT
KLF3 siRNA-656	F:CUUCCAAUGACCUCAUUGUTT R:ACAAUGAGGUCAUUGGAGGTT
siRNA NC	F: UUCUCCGAACGUGUCACGUTT R: ACGUGACACGUUCGGAGAATT
miR-21-5p mimic	UAGCUUAUCAGACUGAUGUUGA
Mimic NC	UUGUACUACACAAAAGUACUG
miR-21-5p inhibitor	UCAACAUCAGUCUGAUAAGCUA
Inhibitor NC	CAGUACUUUUGUGUAGUACAA

**Table 2 genes-12-00814-t002:** Primers for qRT-PCR.

Primer Name	Primer Sequences (5′-3′)	Length (bp)	Accession Number
MyoG	F:CGTGTGCCACAGCCAATG R:CCGCCGGAGAGAGACCTT	63	NM_204184.1
MyHC	F:GAAGGAGACCTCAACGAGATGG R: ATTCAGGTGTCCCAAGTCATCC	138	NM_001319304.1
MyoD	F:GCTACTACACGGAATCACCAAAT R:RCTGGGCTCCACTGTCACTCA	66	NM_204214.2
PCNA	F: AACACTCAGAGCAGAAGAC R: GCACAGGAGATGACAACA	225	NM_204170.2
CDK2	F:GCTCTTCCGTATCTTCCGCA R: ATGCGCTTGTTGGGATCGTA	192	NM_001199857.1
CCND1	F:CTCCTATCAATGCCTCACA R:TCTGCTTCGTCCTCTACA	152	NM_205381.1
MyF5	F:TTCCCTGAGGATTTCGAGCC R:CTCATAGTGGCTGCCTTCCG	197	NM_001030363.1
KLF3	F:CCCCGTTTCAGTGTCATACCC R:TGAGTTTCGCTTGTTCACCG	175	XM_015285673.3
GAPDH	F:GGTGGCCATCAATGATCCCT R:CCGTTCTCAGCCTTGACAGT	105	NM_204305.1

## Data Availability

Not applicable.

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
