# Peer review of "miR-21-5p Regulates the Proliferation and Differentiation of Skeletal Muscle Satellite Cells by Targeting KLF3 in Chicken"

_genes, 2021, doi:10.3390/genes12060814_

Round 1

Reviewer 1 Report

This manuscript demonstrate the role of one microRNA, miR-21-5p into myogenesis in chicken, mainly in culture. The authors identified on target gene of that miR, KLF3. This study is well conducted and convincing. However, I have several remarks which could significantly improve the demonstration and strengthen the data.

Remarks:

1/ Abstract: The 1st sentence is too general and too wide. There plenty papers concerning SMSCs. Please, introduce the paper with less general sentences.

2/ In the introduction, the first paragraph is again too general. It does not bring any interesting indication to the reader.

Furthermore, line 38, what is the difference between myogenic stem cells and SMSC?

Line 45: “microRNAs (miRNAs) are transcribed in long transcripts”. What do you mean by “long transcripts”?

Lines 50-51: “at the post-transcriptional level”. Does it mean that the 3’UTR targeting is not a post-transcriptional regulation?

Lines 63-64: “The effect of 63 miR-21 on skeletal muscle cells has also been reported in many animal models. For in-64 stance, miR-21 is differentially expressed in the skeletal muscle of pigs”. It would be interesting to have further and more precise indications about the differential regulation of this miRNA in pigs.

Line 76: What are the KLF3 subtypes?

Line 81: “its role in skeletal muscle 80 proliferation and differentiation has not been fully understood in animal models includ-81 ing poultry.”. What is known about it?

Overall, a paragraph concerning KLF and more precisely KLF3 and their role in myogenesis needs to be added. The same  for muscle development in chicken.

3/ Figure 1: The correspondence to the different species are not indicated in the legend nor in the text. In figure B, the development stage where the analysis was done is not mentioned. In the panel C, there is no indication concerning the skeletal muscle analyzed. The statistical analyses are not clear (no stars in the figures, but a and b, different from the legend)

4/ Figure 2 is clear. B-tubulin on WB should be presented below the proteins of interest (Figures E and F). Figures L and M, the negative controls mimics and inhibitors should be presented at the left, as in figure A.

5/ Figure 3: same remarks as in figure 2 concerning B-tubulin and negative controls. Which MyHC has been followed by WB and immunostaining? The expression pattern of other MyHC would give important information about the maturation of these myotubes. To assess the differentiation process, the number of nuclei per myotube should be quantified and presented in this figure. On parallel, the percentage of nuclei into myotubes compared to the total nuclei would also add indications about the differentiation status of these cultures. The immunofluorescence images are too small and not clear enough to visualize myotubes. Instead of measuring the myotube area, quantification of their diameter would add some valuable data.

6/ Line 303: “Since many members of the KLFs family play a pivotal role in the proliferation and 303 differentiation of skeletal muscle,”. A reference is missing.

7/ Figure 4:  In B, U instead of T in the mutant sequence. The G panel is not indicated in the legend. B-tubulin should again be presented below the KLF3 panel. On the WB, mimic NC and inhibitor NC conditions are not homogenous: the level of KFL3 expression is not the same, while they are all controls. Is there any technical problem (a transfer problem) ? DF-1 cells were used for these experiments. These results should be confirmed with a skeletal muscle cell line or SMSC. Furthermore, DF-1 cell culture is not described in the material and methods section.

8/ Figure 5: Panels are mixed in the text, and the legends. Figure B is not referenced. As before, B-tubulin should figure out on the bottom of the panel. Expression of KLF3 should also be presented in the WB panel. On this panel, the negative control should figure on the left and the si-KLF3 condition on the right. In panel E and H, the negative control should figure out on the left too. The same remarks for figure 6.

9/ On figure 6, the questions are similar to those of figure 3. The differentiation process should be described in more details. Cf remarks above.

10/ In conclusion, KLF3 inhibits both proliferation and differentiation. So, what could be the role of KLF3 in myogenesis. Does KLF3 regulate cell death ?

11/ line 388: this report does not conclude about the fusion myoblasts into myotubes. This should be discussed or removed from discussion.

Minor points:

  • Line 25: KLF3 was ONE OF THE direct genes … this is probably not THE direct gene of this microRNA
  • Line 62: A Subsequent study reported that miR-21 plays an central role…
  • Line 434: that play a roles in the proliferation

Author Response

1/ Abstract: The 1st sentence is too general and too wide. There plenty papers concerning SMSCs. Please, introduce the paper with less general sentences.

>Response:Thank you for your comments. Instead of general sentence, we have specified the important functions of SMSCs to the proliferation and differentiation of skeletal muscle. (Line 14-15)

2/ In the introduction, the first paragraph is again too general. It does not bring any interesting indication to the reader. Furthermore, line 38, what is the difference between myogenic stem cells and SMSC?

>Response:Thank you for your comments. SMSCs are a heterogeneous population of myogenic stem and progenitor cells that are required for the growth, maintenance and regeneration of skeletal muscle. 

Line 45: “microRNAs (miRNAs) are transcribed in long transcripts”. What do you mean by “long transcripts”?

>Response:Thank you for your comments. We are sorry that there is an error in the language description in our manuscript. microRNAs (miRNAs) are a class of non-coding single-stranded RNA molecules encoded by endogenous genes with a length of about 21 nucleotides, and “Long transcripts” are endogenous genes. We have made revisions to the manuscript. (Line 43-44)

Lines 50-51: “at the post-transcriptional level”. Does it mean that the 3’UTR targeting is not a post-transcriptional regulation?

>Response:Thank you for your comments, we are sorry that the language here is not clear enough. miRNAs regulate genes at the post-transcriptional level, whether binding to 3 'UTR or 5' UTR. And we have revised this sentence to make it easier for readers to read. (Line 45-48)

Lines 63-64: “The effect of 63 miR-21 on skeletal muscle cells has also been reported in many animal models. For in-64 stance, miR-21 is differentially expressed in the skeletal muscle of pigs”. It would be interesting to have further and more precise indications about the differential regulation of this miRNA in pigs.

>Response:Thank you for your comments, we have improved this part in the manuscript. We provide a more detailed description of the differential expression of miRNA during the development of pig skeletal muscle. (Line 60-63)

Line 76: What are the KLF3 subtypes?

>Response:Thank you for your comments. Sequence-tagged transcripts in the GenBank database include several spliceoforms. However, only two KLF3 transcripts were detected during muscle differentiation in this study.

Line 81: “its role in skeletal muscle proliferation and differentiation has not been fully understood in animal models including poultry.”. What is known about it?

>Response:Thank you for your comments. KLF3 is enriched in the MPEX loci of many muscle gene promoters. In addition, KLF3 begins to be expressed late in muscle differentiation, when many of the genes that define mature muscular canals are induced. We have added to the manuscript. (Line 72-73)

Overall, a paragraph concerning KLF and more precisely KLF3 and their role in myogenesis needs to be added. The same  for muscle development in chicken.

>Response:Thank you for your comments. We have added to the manuscript the role of KLFs, KLF3 in myogenesis, and its role in skeletal muscle development in chickens. (Line 72-78)

3/ Figure 1: The correspondence to the different species are not indicated in the legend nor in the text. In figure B, the development stage where the analysis was done is not mentioned. In the panel C, there is no indication concerning the skeletal muscle analyzed. The statistical analyses are not clear (no stars in the figures, but a and b, different from the legend)

>Response:Thank you for your suggestion. We have made revisions to the manuscript. (Line 230-235)

4/ Figure 2 is clear. B-tubulin on WB should be presented below the proteins of interest (Figures E and F). Figures L and M, the negative controls mimics and inhibitors should be presented at the left, as in figure A.

>Response:revised. (Figure 2)

5/ Figure 3: same remarks as in figure 2 concerning B-tubulin and negative controls. Which MyHC has been followed by WB and immunostaining? The expression pattern of other MyHC would give important information about the maturation of these myotubes. To assess the differentiation process, the number of nuclei per myotube should be quantified and presented in this figure. On parallel, the percentage of nuclei into myotubes compared to the total nuclei would also add indications about the differentiation status of these cultures. The immunofluorescence images are too small and not clear enough to visualize myotubes. Instead of measuring the myotube area, quantification of their diameter would add some valuable data

>Response:Thank you for your comments. We used MYHC antibody to include MYH1, MYH10, MYH11, MYH15, MYH2, MYH3, MYH4 and MYH6. The article numbers of the antibodies have been added to the manuscript. In addition, we recalculated the number of nuclei in the myotube and the diameter of the myotube and added it to Figure 3.

6/ Line 303: “Since many members of the KLFs family play a pivotal role in the proliferation and 303 differentiation of skeletal muscle,”. A reference is missing

>Response:Thank you for your comments. We have added references here in the manuscript. (Line 296-297)

7/ Figure 4:  In B, U instead of T in the mutant sequence. The G panel is not indicated in the legend. B-tubulin should again be presented below the KLF3 panel. On the WB, mimic NC and inhibitor NC conditions are not homogenous: the level of KFL3 expression is not the same, while they are all controls. Is there any technical problem (a transfer problem) ? DF-1 cells were used for these experiments. These results should be confirmed with a skeletal muscle cell line or SMSC. Furthermore, DF-1 cell culture is not described in the material and methods section.

>Response:Thank you for your comments. We have corrected the errors in Figure 4. In order to make the KLF3 expression in the mimic NC group and the inhibitor NC group be the same, we reunified the protein concentration and performed WB analysis. In addition, the Luciferase Reporter Assay is susceptible to the difference in the number, uniformity and vitality of the primary cells. However, the DF-1 cell line is a stable cell line after purification, which can avoid experimental errors. We have supplemented the DF-1 cell culture process in the material method. (Line 112-113, Figure 4)

8/ Figure 5: Panels are mixed in the text, and the legends. Figure B is not referenced. As before, B-tubulin should figure out on the bottom of the panel. Expression of KLF3 should also be presented in the WB panel. On this panel, the negative control should figure on the left and the si-KLF3 condition on the right. In panel E and H, the negative control should figure out on the left too. The same remarks for figure 6.

 >Response:Thank you for your suggestion. We have modified the figure 5, and added the WB analysis of the KLF3 knock down. (Line 326-329)

9/ On figure 6, the questions are similar to those of figure 3. The differentiation process should be described in more details. Cf remarks above.

>Response:revised. (Figure 6)

10/ In conclusion, KLF3 inhibits both proliferation and differentiation. So, what could be the role of KLF3 in myogenesis. Does KLF3 regulate cell death ?

>Response:Thank you for your comments. The proliferation and differentiation of muscle cell are important biological processes in skeletal myogenesis. Our results suggest that KLF3 has a negative effect on proliferation and differentiation of chicken SMSCs, and therefore may affect muscle development by inhibiting the expression of some muscle-derived genes. Qing Mao et al. found that Transfection of KLF3-AS1 exosome in rats and incubation with KLF3-AS1 exosome in hypoxia cardiomyocytes bothVerified that overexpression of KLF3-AS1 in exosomes leads to reduced MI area, decreased cell apoptosis. However, we did not find any papers on how KLF3 affects skeletal muscle cell death.

11/ line 388: this report does not conclude about the fusion myoblasts into myotubes. This should be discussed or removed from discussion.

>Response:Thank you for your comments. We have revised and rediscussed this part of the manuscript. (Line 371-374)

Minor points:

Line 25: KLF3 was ONE OF THE direct genes … this is probably not THE direct gene of this microRNA

>Response:revised. (Line 24)

Line 62: A Subsequent study reported that miR-21 plays an central role…

>Response:revised. (Line 57)

Line 434: that play a roles in the proliferation

>Response:revised. (Line 423)

Reviewer 2 Report

This is very well written and very interesting manuscript deciphering the role of miR-21-5p in chicken skeletal muscle development. There are few minor discrepancies that need to be addressed:

  1. Line 113: you may consider renaming DM1, DM2, DM3 and DM4 using the time of collection instead 1, 2, 3, etc. I would suggest: DM24, DM36, DM48 and DM72.
  2. Line 124: please revise, replace word “updated”
  3. Lines130-146: how much RNA was used for each cDNA reaction?
  4. Lines 149-159: how much protein was loaded in each well? What was the source and dilution of secondary antibody? How was the signal developed?
  5. Figure 1: please define abbreviation for species (panel A), logically, first should be panel D and then panel C.
  6. All figures: figures in current form are too small, the text is too small and need to be enlarged, at least 50%. Please consider splitting Figures into couple of panels only, or the figures should be full page.
  7. Immunofluorescent figures need to be enlarged.

Author Response

Line 113: you may consider renaming DM1, DM2, DM3 and DM4 using the time of collection instead 1, 2, 3, etc. I would suggest: DM24, DM36, DM48 and DM72.

>Response:Thank you for your suggestion. We have revised this part of the manuscript. (Line 112)

Line 124: please revise, replace word “updated”

>Response:Thank you for your comments. We've replaced "updated" with "replaced". (Line 124)

Lines130-146: how much RNA was used for each cDNA reaction?

>Response:Thank you for your comments. In our experiment, the reverse transcripts were all 10ul systems, in which 1ulRNA samples were used for cDNA reverse transcripts of mRNAs, while 3.75ul was needed for miRNA.

Lines 149-159: how much protein was loaded in each well? What was the source and dilution of secondary antibody? How was the signal developed?

>Response:Thank you for your comments. The proteins were cleaved according to the instructions of the Total Protein Extraction Kit, using 200ul of lysate per well of the six-well plate. The secondary antibody are as follows: HRP Goat Anti-Rabbit IgG (H+L) (ABclonal, Wuhan, China; 1:5000); Goat Anti-mouse IgG (Biorbyt, Cambridge, United Kingdom; diluted, 1:5000), and we have made a supplement in the manuscript. Because Western Blot assays use the methods of Cui et al., we didn't state the details. the electrochemiluminescence (ECL) method was used to develop, and the photographswere obtained using ChemiDoc™ MP Imaging System. (Line 157-159)

Figure 1: please define abbreviation for species (panel A), logically, first should be panel D and then panel C.

>Response:Thank you for your comments. We have defined the abbreviation for species in panel A and swapped the positions of panels C and D and of figures C and D.

All figures: figures in current form are too small, the text is too small and need to be enlarged, at least 50%. Please consider splitting Figures into couple of panels only, or the figures should be full page.

>Response:revised.

Immunofluorescent figures need to be enlarged.

>Response:revised.

Round 2

Reviewer 1 Report

First I would like to thank the authors for their answer and to have completed their work. I still have few remarks concerning the manuscript:

1/ The difference between myogenic stem cells and SMSC has to be clearly defined in the manuscript. In mammals, there is no real difference between them.  Satellite cells are muscle stem cells. Myogenic progenitor cells are then called myoblasts. If it is different in chicken, this should clearly be indicated.

2/ In figure 1B, it is still not clear at which developmental stage the expression of these miRNAs has been analysed.

3/ In Figure 1D, it is still not clear what a, b, and c correspond to.

4/ The panel G is still not mentioned in the legend of figure 4.

 5/ The choice of DF-1 cells rather than primary cells should be mentioned in the manuscript. The type of cells should also be mentioned. If the cells used here are stable cell lines, it should also be described in the text and material and methods.

This manuscript is a resubmission of an earlier submission. The following is a list of the peer review reports and author responses from that submission.

Round 1

Reviewer 1 Report

In this manuscript, entitle “miR-21-5p regulates the proliferation and differentiation of skeletal muscle satellite cells by targeting KLF3 in Chicken”, the authors have conducted an analysis of the mechanism by which miR-21-5p regulates SMSCs. Globally, their study is well conduct and is well written (the manuscript is easy to follow). To my concern, there is some major comments that required explanation to allow a good understanding of the subject by the readers.

Major Comments:

1/ My first concern is about the preliminary sequencing data that the authors mentioned all across the manuscript, are they already published elsewhere? If yes, can this could be more explained and can the authors provide the associated details? Furthermore, if the results are already published in another article, the associated data of gene/miR expression should not appear twice in two different articles.

2/ In the material and methods section, it is not clear how many animals were used to get SMSCs.

3/ In the material and methods section, the authors did not detailed how the differenciation efficiency is measured. Is it through the markers MyoG and MyHC expression? If yes, can the authors add reference to justify better and provide their expression rate following the different states of differenciation?

4/ The authors used U6 as reference miR for miR quantification. The authors should state how U6 is  stable between samples and between different experimental conditions.

5/Line 207, in the statistical analysis part, the authors did not mentioned how they conducted their statistics between two conditions like in figure 2.

6/ Figure 2, panels J and 2K : it seems that there is differences between mimic NC and inhibitor NC conditions in terms of number of proliferatives cells. Did the authors cheked this comparision in details ? Furthermore, did they made this comparision with untransfectd cells ?

7/ A general comment for the entire manuscript : for each figure, it is not clear if the authors have made the experiments 3 times independently and how many biological replicates and technical replicates they did. Thank you to clarify this point.

8/ Figure 3, panels B and C : the authors have to specified if cells are in GM or at what time after the differenciation the experiment was conducted.

9/ Figure 3I : why the authors did not showed the results for MyF5 and MyoD ?

10/ Figure 3J : same comment as in 6/

11/ Line 293, for the the choose of KLF3. How was the expression of this gene in the previous RNAseq analysis ? What about the 15 other genes ?

12/ Figure 5A : some tissues are missing for KLF3 expression compared to miR21 in Figure 1D. Furthermore, can the autors conserved the same order in the grahs 1D and 5A to facilitate the reading ?

13/ There is differencies for the relative myotube area for inbibitor NC (figure 3H), mimic NC (figure 3J) and si-NC (figure 6F), can the authors explain this ? This 3 experiments have been made all in the same time ?

14/ Did the authors tried to measure KLF3 expression following miR-21 inhibition in the differentiate states ?

15/ Instead of providing too many informations on the human’s litterature, more details would be appreciated on what is known and what have been already studied in microRNAs regulation and more especially myomiRs regulation in chicken.

16/ Please add « in chicken » line 394. Furthermore, a better organised discussion in term of type of cell/organism/patologies would be easier to read.

17/ In the introduction and discussion, it is laking details on how this study could be interesting in the agronomic area, for meat-producing chicken for example.

Minor Comments:

1/ In the introduction, as well as in the discussion, the authors state that “the skeletal muscle is the largest organ in the human body” but as far as I know, it is the skin.

2/ The authors should clarify better if skeletal muscle satellite cells are primary cells and how many passages are they keeping them in culture.

3/ What is the diet used for the animals in this study?

4/ Figure 1 in the legend (line 232), NC for negative control is stipulated but there is no NC condition in the figure. Furthermore, C and D panels should be reversed.

5/ Figure 2H: can the authors make the squares and dots more visible.

6/ Figures 2L and 2M: it is confusing as it is not the same color as it was used in 2A, 2B, 2C, 2D.

7/ In vitro should be written in italics.

8/ Figure 4E : please clarify the n number.

9/ The panel F in the Figure 5 should be next to the E.

10/ In the dicussion, line 435, the authors cite a article in which KLF3 seems to inhibate Wee1, did they checked its expression in their samples ?

Reviewer 2 Report

The work is quite chaotic and hard to read. Many important elements are missing from such a large number of analyzes. I was also disappointed that all the results were not discussed. What raises my biggest concerns is too low number of individuals included in the analysis.

line 16-17 - In this study, we studied.. please correct this sentence
line 31- do not duplicate keywords with the title of the publication
line 34- starting with a sentence about people at work about chickens is not accurate
line 37- double "up"
line 43- indicate this role
line 46 - again "important role in numerous biological processes" indicate a specific role and processes, try to avoid the frequent use of the word "important" without specifying what it means
line 53-54- Rearrange the sentence, it doesn't sound right. For the reader, the perception is that muscle-associated miRs are expressed in muscle.
line 58-62- a sentence unrelated to the topic of work

line 94- 3 biological repetitions is definitely not enough for this type of analysis. In biological tests, a minimum of 5-6 biological repetitions is assumed. The data presented on such a small number are not reliable.
line 97- the purpose for which these tissues were collected was not indicated
line 99 - previous studies? it was not indicated in which one.
line 100- how many birds were used for this procedure
line 104 - inconsistent administration of products - you indicate the place of origin once, sometimes only the company
line 112- biological or technical?
line 115- double "mimic"
line 116- the abbreviation was already introduced earlier
Table 1 - the table is not correct - lines have shifted
line 139- what means "related genes"
line 140- 3 biological repetitions is definitely not enough, especially when it comes to tissues. Too much diversity between individuals does not allow such a conservative analysis. How many technical repetitions ???
line 141- why were these genes selected as reference?
line 144- extraction from what tissue? how many repetitions?
Table 2- the same what in table 1; where did the starters come from? how are they designed?;
the sequence on which the primers were designed was also not indicated. what base did she come from?
line 206- was the distribution consistent with normal? which test was used? what about the homogeneity of the variance?
In this case, it is difficult to talk about a reliable statistical analysis with such a small number.
You write about different embryo development stages- but in M&M only about tissues from 3days-old chickens?? could you explain? 
Figure 1- you indicate the thresholds of statistical significance in the description of the figure, but there is no sign in the graph as indicated in the description.
The abbreviations "a,b,c" are also given without any indication of what they mean. (D) The 10 miRNAs with the highest expres-230 sion in embryonic chicken skeletal muscle - but I see only different tissues not miR.
line 366-repeating the sentence from the introduction not related to the topic
line 372- these lines are a repetition of the introduction - re-paraphrased sentences that have already appeared before
The discussion is too short, it does not discuss the obtained results, which are so numerous in the manuscript. It duplicates the fragments that were already mentioned in the introduction.
There are also too many repetitions of methodologies and results in the discussion.
It does not indicate new scientific knowledge emerging from this work.